# α-Terpineol Induces Shelterin Components TRF1 and TRF2 to Mitigate Senescence and Telomere Integrity Loss via A Telomerase-Independent Pathway

**DOI:** 10.3390/antiox13101258

**Published:** 2024-10-17

**Authors:** Marianna Kapetanou, Sophia Athanasopoulou, Andreas Goutas, Dimitra Makatsori, Varvara Trachana, Efstathios Gonos

**Affiliations:** 1Institute of Chemical Biology, National Hellenic Research Foundation, 11635 Athens, Greece; mkapetanou@eie.gr (M.K.); sathan@eie.gr (S.A.); 2Hellenic Pasteur Institute, 11521 Athens, Greece; dmak@pasteur.gr; 3Department of Biology, Faculty of Medicine, School of Health Sciences, University of Thessaly, 41500 Larisa, Greece

**Keywords:** senescence, aging, telomeres, shelterin, oxidative DNA damage, antioxidants, PI3K/AKT pathway

## Abstract

Cellular senescence is a hallmark of aging characterized by irreversible growth arrest and functional decline. Progressive telomeric DNA shortening in dividing somatic cells, programmed during development, leads to critically short telomeres that trigger replicative senescence and thereby contribute to aging. Therefore, protecting telomeres from DNA damage is essential in order to avoid entry into senescence and organismal aging. In several organisms, including mammals, telomeres are protected by a protein complex named shelterin that prevents DNA damage at the chromosome ends through the specific function of its subunits. Here, we reveal that the nuclear protein levels of shelterin components TRF1 and TRF2 decline in fibroblasts reaching senescence. Notably, we identify α-terpineol as an activator that effectively enhances TRF1 and TRF2 levels in a telomerase-independent manner, counteracting the senescence-associated decline in these crucial proteins. Moreover, α-terpineol ameliorates the cells’ response to oxidative DNA damage, particularly at the telomeric regions, thus preserving telomere length and delaying senescence. More importantly, our findings reveal the significance of the PI3K/AKT pathway in the regulation of shelterin components responsible for preserving telomere integrity. In conclusion, this study deepens our understanding of the molecular pathways involved in senescence-associated telomere dysfunction and highlights the potential of shelterin components to serve as targets of therapeutic interventions, aimed at promoting healthy aging and combating age-related diseases.

## 1. Introduction

Cellular senescence, a state of irreversible growth arrest accompanied by alterations in cellular morphology and function, plays a significant role in driving age-related pathologies and functional decline [1,2,3]. At the molecular level, cellular senescence is intricately linked to the integrity of telomeres, specialized nucleoprotein structures that cap the ends of linear chromosomes. Telomeres protect genomic DNA from degradation, prevent chromosomal fusions and mitigate the activation of DNA damage response pathways. The latter is crucial as, due to the end-replication problem, telomeres undergo progressive shortening with each round of cell division in somatic cells; this gradual erosion of telomeric DNA eventually leads to the exposure of chromosome ends, triggering a DNA damage response that will eventually lead to cellular senescence [1]. Importantly, short telomeres have been identified as an independent risk factor for functional decline in elderly populations in Europe [4].

The linear arrangement of eukaryotic chromosomes presents significant challenges to genome integrity. Firstly, the conventional replication machinery’s inability to fully replicate parental DNA termini leads to the inevitable erosion of chromosome ends, a phenomenon compensated either by telomerase-dependent telomere elongation or Alternative Lengthening of Telomeres (ALT) in germ, stem and cancer cells [5]. Secondly, DNA extremities are susceptible to misidentification as damage by the DNA damage response (DDR) machinery, potentially triggering cellular senescence, apoptosis or double-strand break repair processes [6]. Telomerase, a ribonucleoprotein enzyme complex, plays a pivotal role in maintaining telomere length by adding repetitive DNA sequences to chromosome ends, thereby counteracting progressive telomere shortening. Even though telomerase has the potential to serve as a prime therapeutic target, limitations in telomerase-based strategies, including the consequential activation of ALT-driven telomere elongation or possible pro-aging or prooncogenic effects, have spurred the development of alternative telomerase-independent approaches [7]. Targeting the complexes forming telomeres themselves could thus offer complementary or alternative routes for telomere-based therapies.

In addition to replicative attrition, telomeres are sensitive to various endogenous and environmental factors, such as alcohol, caffeine, heat shock, stress hormones, improper cell cycle progression through mitosis and oxidative or other genotoxic stress [8]. Oxidative stress, in particular, has been implicated in accelerating telomere shortening and dysfunction through the accumulation of oxidative DNA damage, including the formation of 8-oxoG, in guanine (G)-rich telomeric regions. Short telomeres are particularly sensitive to oxidative damage, as they are prone to DNA breakage and dysfunction, potentially leading to genomic instability [9]. Furthermore, reactive oxygen species (ROS), in addition to inducing myriad DNA lesions, alter cell signaling and gene expression, indirectly affecting telomere biology [10,11]. Consequently, telomeres have emerged as vital regulators of the cell cycle, senescence and lifespan, with experimental and pathological evidence linking telomere dysfunction to genome-wide deleterious consequences and oncogenesis [7,12].

Hence, protecting telomeres from the DNA damage response is essential to circumvent cellular senescence, organismal aging and age-related diseases. In numerous organisms, including mammals, telomeres are safeguarded by a protein complex named shelterin, which counteracts the DNA damage response at the chromosome ends through the specific function of its subunits. The modulation of shelterin structure and function during development and aging constitutes an intense area of research.

The shelterin protein complex, consisting of six core subunits—TRF1, TRF2, POT1, TIN2, TPP1 and RAP1—plays a central role in maintaining telomere integrity and function. Shelterin proteins collectively regulate telomere length, protect chromosome ends from inappropriate DNA repair activities and modulate telomere dynamics during cell cycle progression and cellular stress responses. Among the shelterin components, TRF1 and TRF2 (telomeric repeat-binding factor 1 and 2) are particularly critical for telomere protection. TRF1 binds to the canonical double-stranded telomeric repeats (TTAGGGs). TRF1 binding to telomeric DNA is essential for the protection and stability of telomeres. It helps to prevent telomere erosion, inhibits DNA damage responses and regulates telomere length. The absence of TRF1 elevates the risk of telomere dysfunction, leading to genome instability and eventually senescence or apoptosis. TRF2 has a distinct role that facilitates the formation of a protective T-loop structure in the telomeres, shielding them from DNA damage repair machinery while preventing end-to-end chromosome fusions [13]. Both TRF1 and TRF2 contribute to the regulation of telomere length and telomerase activity and therefore their dysregulation can lead to telomere dysfunction and genomic instability, affecting senescence and organismal aging [14].

In this context, there is a compelling need for novel therapeutic interventions targeting telomere dysfunction to promote healthy aging and combat age-related diseases [15]. Thus, modulation of shelterin components, particularly TRF1 and TRF2, emerges as an enticing avenue for scientific exploration. To this end, this study explores the effects of an activator capable of modulating TRF1 and TRF2 levels and function, while also examining its ability to safeguard telomere length, with the goal of mitigating senescence-associated telomere attrition and enhancing genomic stability.

## 2. Materials and Methods

### 2.1. Cell Cultures

HFL-1 human diploid fibroblasts (HDFs) were cultured in Dulbecco’s modified Eagle’s medium (Thermo Fisher Scientific, Waltham, MA, USA), which was enriched with 10% (*v*/*v*) fetal bovine serum (Thermo Fisher Scientific, Waltham, MA, USA), 100 units/mL penicillin, 100 μg/mL streptomycin, 2 mM glutamine and 1% (*v*/*v*) non-essential amino acids (complete medium). Cells were maintained in an incubator set at 37 °C, with 5% CO2 and 95% humidity. The culture medium was consistently supplemented with the identified optimal concentrations of compounds dissolved in DMSO (dimethyl sulfoxide). α-Terpineol (catalog no. 04899) was purchased from Sigma-Aldrich (St. Louis, MO, USA). Control cultures were treated with a medium containing 0.1% DMSO. Every 72 h, the media were refreshed with new media containing the compounds or diluent, and cell counts were performed using a Coulter Z2 counter (Beckman Coulter, Brea, CA, USA) until the cells became senescent (around 13 weeks). Media were changed 16 h before any assays. Cumulative population doublings (CPDs) for each culture were determined using the formula: CPD = Σ(PD), where PD = LOG(Nfinal/Ninitial)/LOG [16]. Here, Nfinal is the number of cells at confluence and Ninitial is the initial seeding count.

### 2.2. Induction of Oxidative Stress

One hundred and fifty thousand (150,000) HDFs were seeded per well on a 6-well plate, and at 65–70% confluence, cells were treated with the compounds or DMSO (solvent control) for 24 h. Subsequently, cells were exposed to 300 µΜ H2O2 for 1.5 h in the presence of the compounds or solvent control and were then washed thoroughly with PBS. Treated cultures were left to recover in complete medium for 48 h. Cell survival was assessed by counting the number of cells using a Coulter Z2 counter (Beckman Coulter, Brea, CA, USA) after the recovery period.

### 2.3. Relative Telomere Length

Cells were harvested after treatment with either the compounds or the diluent during their lifespan or following the induction of oxidative stress. DNA was extracted using the NUCLEOSPIN extraction kit (Macherey-Nagel, Düren, Germany). The concentration of DNA was assessed by measuring absorbance at 260 nm with a UV-VIS spectrophotometer and adjusting for the dilution factor. Relative telomere length was evaluated by quantitative PCR following the protocol of Cawthon [17], using the CFX Connect Real-Time PCR Detection System (Bio-Rad Laboratories, Hercules, CA, USA). The relative telomere lengths (T/S ratios) were determined through monochrome multiplex quantitative PCR (MMQPCR), using a set of primers specific to a single-copy gene (albumin, S) and a telomere-specific primer set (T).

### 2.4. Quantification of Telomerase Activity

A real-time PCR-based telomeric repeat amplification protocol (TRAP) assay was performed in HDFs grown in media supplemented with 0,5 μg/mL α-terpineol or the diluent DMSO (control) for 48 h using the TRAPEZE^®^ RT Telomerase Detection Kit (S7710, Sigma-Aldrich, St. Louis, MO, USA), according to the manufacturer’s protocol. The telomerase activity of each sample was calculated as a percentage of the relative telomerase activity of the positive control cells.

### 2.5. Immunoblot Analysis

Nuclear protein extraction was performed using the Nuclear Extraction Kit (ab113474, Abcam, Cambridge, UK) according to the manufacturer’s guidelines. The cytoplasmic protein fraction was also isolated for downstream applications. A total of 20 μg of isolated protein was separated by 10% SDS-PAGE under non-reducing conditions according to standard procedures [18]. Following electrophoresis, protein loading was analyzed using Stain-freeTM (Bio-Rad, Hercules, CA, USA) imaging technology, and the protein fractions were transferred to a nitrocellulose membrane to be probed for TRF1 (ab129177, Abcam, Cambridge, UK) and TRF2 (ab108997, Abcam, Cambridge, UK) or PI 3 Kinase p85 alpha (ab191606, Abcam, Cambridge, UK), PI 3 Kinase p85 alpha phospho Y607 (ab182651, Abcam, Cambridge, UK), AKT1 + AKT2 + AKT3 (ab179463, Abcam, Cambridge, UK) and AKT1 + AKT2 + AKT3 phospho S472 + S473 + S474 (ab192623, Abcam, Cambridge, UK). The antibody for H2A.X was obtained by Upstate Biotechnology (Lake Placid, NY, USA). The mouse monoclonal anti-γH2AX antibody (05–636; phosphor S139, clone JBW301) was obtained from Millipore (St. Louis, MO, USA), while the mouse monoclonal antibodies for p53 (sc-126; DO-1), p16 (sc-1661; F-12) and p21 (sc-397; F-5) were obtained from Santa Cruz Biotechnology (Dallas, TX, USA). Secondary antibodies (ab205718; ab6728, Abcam, Cambridge, UK) linked to horseradish peroxidase were used to visualize the primary antibodies. Detection was performed via enhanced chemiluminescence, utilizing the Clarity ECL Substrate, and imaging was carried out with the Chemidoc XRS + system (Bio-Rad Laboratories, Hercules, CA, USA).

### 2.6. Oxidized Protein Levels

Protein carbonyl groups in cell lysates were detected using the “Protein Carbonyl ELISA Kit” (ab238536; Abcam, Cambridge, UK), according to the manufacturer’s instructions. All measurements were performed in triplicate. Absorbance readings were taken using the Safire II microplate reader (TECAN, Männedorf, Switzerland), and the protein carbonyl content of unknown samples was calculated based on a standard curve generated from BSA standards.

### 2.7. Detection of Oxidized TRF1

Nuclear extracts were prepared as described above and samples were precleared with protein A-agarose beads (sc-2001) for 3 h min at 4 °C. For immunoprecipitation, 500 μg of precleared nuclear protein extract was incubated with 4 μg of TRF1 antibody (ab129177, Abcam, Cambridge, UK) pre-coupled with the protein A-agarose beads, with constant rocking at 4 °C overnight. Immunoprecipitated protein complexes were collected, washed four times in extraction buffer and eluted from the agarose beads by boiling for 5 min in non-reducing Laemmli buffer. Immunoprecipitated proteins at a ratio of 1:20 were analyzed with an OxyBlot™ Protein Oxidation Detection Kit (OxyBlot, Chemicon Cat. no. S7150-Kit) according to the manufacturer’s instructions (Millipore, Billerica, MA, USA).

### 2.8. Immunofluorescence

Early-passage fibroblasts were grown in coverslips, treated with α-terpineol or DMSO (solvent control) for 24 h and then subjected to the above-described oxidative treatment. Immunofluorescence was then performed as previously described [19]. Specifically, cells were fixed in 4% paraformaldehyde followed by cell permeabilization with 0.2% Triton X-100 in PBS. Fixed samples were incubated with primary antibodies of interest (γ-H2AX, 53BP1 or TRF2) at a 1:200 dilution in PBS containing 0.2% Triton X-100 and the appropriate secondary antibodies. Mouse monoclonal anti-γH2AX (05–636; phosphor S139, clone JBW301) and anti-53BP1 (clone BP13,) were purchased from Millipore (MA, USA). The secondary antibodies Alexa Fluor 594 anti-rabbit IgG (A11005) and AlexaFluor 488 anti-mouse IgG (A11001) were obtained from Molecular Probes (Invitrogen). Coverslips were embedded in 10 µL of Anti-Fade with DAPI (4,6-diamidino-2-phenylindole) mounting medium (ab188804, Abcam, Cambridge, UK) and analyzed on a ZEISS Axio Imager Z2 fluorescent microscope. Images were captured on a confocal microscope ZEISS LSM780 using the ZEN 2011 program. For calculations of damaged nuclei, at least 100 cells from at least 5 randomly selected fields were analyzed for each culture condition. Cells with >5 γ-H2AX or 53BP1 foci in their nuclei were counted as positive for DNA damage by a single observer blinded to treatment regimen. For calculations of telomere damage foci, cells with >10 γ-H2AX and TRF2 co-localization points were counted as positive. For readability, data are given as the percent of positive cells.

### 2.9. Inhibition of PI3K/AKT Pathway

One hundred and fifty thousand (150,000) HDFs were seeded per well on a 6-well plate, and at 65–70% confluence, cells were incubated with a final concentration of 10 μM of the pan-Akt inhibitor Miransertib (ab235550, Abcam, Cambridge, UK) and 0.5 μg/mL (3.24 μΜ) α-terpineol or DMSO (solvent control) for 24 h.

## 3. Results

### 3.1. Induction of the Shelterin Components TRF1 and TRF2 by an Activator Enhances Cellular Lifespan and Preserves Telomere Length During Senescence

First, we screened a library of compounds with putative anti-aging properties, and we identified a monoterpene, namely α-terpineol (Figure 1A). We assessed the replicative lifespan of human embryonic fibroblasts treated with varying concentrations of α-terpineol, ranging from 0.2 to 5 μg/mL (1.2–32.4 μΜ). The cells were cultured for a total of 95 days. Notably, cells treated with the compound at a concentration of 0.5 μg/mL (3.24 μΜ) exhibited a moderate statistically significant extension in replicative lifespan compared to their control counterparts (Figure 1B). Specifically, the control cells reached a cumulative population doubling (CPD) of 57, while the terpineol-treated cells reached a CPD of 58.9. Senescent cells expressed the senescence markers p16, p21 and p53, further confirming their senescent state (Figure 1C).

Next, we found that administration of α-terpineol has a remarkable capacity to attenuate telomere attrition. Specifically, fibroblasts treated with 0.5 μg/mL (or 3.24 μΜ) α-terpineol throughout their lifespan exhibited telomeres maintained at nearly 100% of the initial T/S ratio, contrasting sharply with senescent control cells, which exhibited a 75% reduction in the initial T/S ratio (Figure 2A). Importantly, we also revealed that this effect of α-terpineol on telomeres is independent of telomerase activity (Figure 2B). Notably, we also observed a significant reduction in the nuclear protein levels of TRF1 (−45%) and TRF2 (−79%) in senescent fibroblasts (Figure 2C) treated with solvent control. On the contrary, the administration of α-terpinol preserves the levels of TRF1/2 factors in the nucleus in senescent cells independently of telomerase activity.

### 3.2. Treatment with α-Terpineol Reduces Oxidative Stress-Induced DNA Damage

As α-terpineol induces TRF1/2 factors, which are crucial in safeguarding telomeric regions and consequently shielding DNA from damage, our subsequent aim was to explore its possible contribution to mitigating oxidative DNA damage. As demonstrated in Figure 3A, oxidative stress compromised cellular viability, and α-terpineol offered significant protection, as indicated by enhanced cellular survival numbers after treatment with H_2_O_2_. To assess the effects on DNA integrity, we employed two DNA damage markers, γH2αX and 53BP1. Intriguingly, administration of α-terpineol not only decreased baseline levels of DNA damage from 34.0 ± 0.9% to 23.3 ± 1.9% (as evidenced by γH2AX) in fibroblasts treated for 24 h with α-terpineol but also significantly reduced oxidative stress-induced DNA damage (83.0 ± 1.4% to 55.7 ± 2.7% nuclei positive for γH2AX in the presence of α-terpineol). This indicates that α-terpineol effectively counteracts oxidative damage to DNA (Figure 3B). Similar results were obtained when examining the second DNA damage marker, 53BP1, further validating our afore-mentioned observations considering the DNA-protective role of α-terpineol (Appendix A).

### 3.3. α-Terpineol Protects from Oxidative Stress-Induced Telomere-Specific DNA Damage

Preserving DNA integrity from oxidative damage holds particular significance in telomeric regions, where maintaining genomic stability is critical for preventing senescence. To identify oxidative damage within telomeric regions, also known as telomere dysfunction-induced foci (TIFs), we examined the co-localization of the DNA damage marker γH2AX with TRF2. Our analysis, presented in Figure 4, revealed that, even in cells not exposed to exogenous oxidative stress, α-terpineol reduced the percentage of cells exhibiting TIFs from 11.9 ± 1.0% to 7.1 ± 1.0%. Similarly, α-terpineol shielded telomeric regions from oxidative damage. Specifically, treatment with H_2_O_2_ increased the percentage of cells exhibiting TIFs to 36.7%, while only 27.1% of α-terpineol-treated cells showed TIFs. Therefore, α-terpineol emerges as a promising candidate for safeguarding DNA integrity within telomeric regions against oxidative stress.

### 3.4. α-Terpineol Reduces Oxidative Damage of Nuclear Proteins and Shelterin Components During Senescence

To further investigate the potential antioxidant actions of α-terpineol, next, we assessed the levels of oxidized proteins in both early-passage and senescent cells treated with either α-terpineol or the solvent (control). As shown in Figure 5 regarding total proteins, no differences were observed between cells treated with the compound and control; in both cases, oxidized proteins doubled during senescence. However, when examining nuclear proteins, α-terpineol exhibited prominent antioxidant effects in both early-passage (−58%) and senescent (−30%) cells (see Figure 5B). Notably, α-terpineol significantly reduced the levels of oxidized TRF1 in senescent cells compared to their respective control cultures (Figure 5C). These findings strongly indicate that α-terpineol selectively exerts its antioxidant effects within the nuclear environment.

### 3.5. PI3K/AKT Signaling Mediates the Effects of α-Terpineol in Shelterin Components

It was recently reported that PI3K/AKT inhibitors diminished TRF1 telomeric foci while increasing telomeric DNA damage [20]. In this context, we aimed to explore whether the maintenance of TRF1/2 levels and function of a-terpineol reported here could be mediated by the PI3K/AKT signaling pathway. Interestingly, treatment of fibroblasts with α-terpineol for 24 h led to a notable increase in phosphor-AKT levels, suggesting activation of the PI3K/AKT pathway (Figure 6A). To further explore the role of this pathway in α-terpineol-induced TRF1/2 expression, we utilized the chemical pan-AKT inhibitor miransertib. Strikingly, inhibition of AKT phosphorylation abolished the effects of α-terpineol in TRF1/2 level maintenance, indicating that the PI3K/AKT pathway mediates the upregulation of TRF1/2 by this compound (Figure 6B). Taken together, our results indicate that the effects of α-terpineol on shelterin components, telomere maintenance and possibly its anti-aging abilities are mediated by PI3K/AKT signaling.

## 4. Discussion

The crucial role of TRF1 and TRF2 in safeguarding telomeres underscores their significance in genomic stability and cellular aging [21,22]. Our study reveals a notable decrease in the nuclear protein levels of TRF1 and TRF2 in senescent fibroblasts, indicating their implication in telomere integrity loss that contributes to senescence. The senescence-associated decline in their levels could be responsible for a collapse of telomeric capping and subsequent DNA damage, as previously reported [22,23,24]. This notion is also supported by studies demonstrating that genetic deletion or inhibition of TRF1 results in telomeric DNA damage, decreased proliferation and apoptosis [25]. Similarly, it has been reported that reduced expression of TRF1 in aged endothelial cells [26] and of TRF2 in senescent vascular smooth muscle cells results in compromised DNA damage response and cellular senescence [27]. Furthermore, TRF1 is essential for the induction and maintenance of pluripotency [28] and age-related reduction in TRF1 levels in human and murine tissues, further emphasizing its involvement in organismal aging [29].

Despite the significant advancements in the field of telomere biology, there is a notable absence of known activators that would be capable of directly regulating TRF1 or TRF2. Remarkably, our study introduces a promising avenue for addressing this limitation by demonstrating the ability of α-terpineol to directly increase the nuclear levels of these shelterin components and maintain these high levels upon senescence. This could be the reason for telomere length preservation in α-terpineol-treated senescent cells that were proved here to be independent of telomerase activity. This intriguing outcome implies that, despite the observed decrease in telomerase activity, α-terpineol promotes alternative mechanisms that protect or stabilize telomeres through pathways such as reduced telomere erosion. In contrast, if the telomerase RNA component (Terc) was inhibited, one would expect a more significant reduction in telomere length due to the impaired synthesis of telomeric repeats. The latter converts this activator into a highly promising anti-aging factor, as it prevents potential disruption of tumor suppressor pathways.

Furthermore, as we demonstrated here, the elevated levels of TRF1 and TRF2 induced by α-terpineol create a robust defense against oxidative DNA damage. Telomeric DNA, composed of TTAGGGs, is guanine-rich, rendering it highly prone to oxidation due to guanine’s low redox potential [10,30]. Oxidative damage significantly impacts telomere integrity by leading to over 50% reduced binding of TRF1 and TRF2, with just one oxidized guanine within the telomeric tract [31]. Consequently, this decrease in TRF1 and TRF2 binding could lead to telomere uncapping and therefore predispose cells exposed to oxidative stress to senescence [32,33]. Our results here describe a TRF1/2 activator that mitigates oxidative DNA damage, particularly in telomeric regions, which is of crucial importance for maintaining genomic stability. As shown, α-terpineol not only reduces basal levels of DNA damage but also significantly diminishes oxidative stress-induced DNA damage, as indicated by decreased γH2AX levels after exogenous oxidative insult, highlighting its effectiveness in counteracting oxidative damage to DNA. Furthermore, when the effect of this activator on oxidative damage protection within the telomeric regions, referred to as telomere dysfunction-induced foci (TIFs) [34], was explored, α-terpineol not only reduced the percentage of cells exhibiting TIFs under normal conditions but also shielded telomeric regions from oxidative damage induced by H_2_O_2_ treatment. These findings align with previous studies that have highlighted the crucial role of TRF1/2 in safeguarding telomeric DNA integrity and shielding it from oxidative damage [21]. Together, our results emphasize the potential of TRF1/2-activating compounds, like α-terpineol, as promising candidates for preserving genomic stability, particularly within telomeric regions, against oxidative stress-induced DNA damage.

Our investigation into the antioxidant actions of α-terpineol extended to exploring its impact on proteostasis. Remarkably, α-terpineol’s antioxidant capacity does not demonstrate a ubiquitous effect. Instead, our findings reveal a selective protection of nuclear proteins from oxidative damage and, specifically, of shelterin components like TRF1 upon cellular senescence. A-Terpineol did not show a generalized antioxidant capacity for cytoplasmic proteins. Furthermore, our observations also revealed that oxidized TRF1 accumulates in the nucleus of senescent fibroblasts, possibly further compromising telomere integrity and accelerating cellular senescence. These findings underscore the importance of understanding the specific targets and exact mechanisms of action of different antioxidant compounds. This is extremely important as it challenges the attractive “one-size-fits-all” approach and emphasizes the need to investigate the precise mechanisms of action of the highly appealing and increasingly popular numerous antioxidant agents [35]. In this context, previous studies by our group have identified a set of compounds that act as “secondary antioxidant scavengers” by directly activating the proteasome, delaying aging as well as the progression of age-related diseases [36,37,38,39].

The regulation of telomeric proteins by non-telomeric signaling pathways adds a layer of complexity to the intricate interplay between telomere maintenance and cellular signaling. For instance, TRF2 expression modulation has been linked to the Wnt/β-catenin pathway, a signaling cascade pivotal in stem cell pluripotency, embryonic development and tumorigenesis [40]. Similarly, recent studies have unveiled the regulatory role of PI3Kα, a downstream effector of AKT, in modulating TRF1 protein expression [20]. Specifically, PI3K/AKT inhibitors decreased TRF1 telomeric foci and increased telomeric DNA damage. AKT-mediated phosphorylation of TRF1 regulated its stability and telomeric DNA binding. Breast cancer models treated with a PI3Kα-specific inhibitor showed reduced TRF1 levels and increased DNA damage, indicating a connection between telomeres, the PI3K pathway and cancer progression [20]. In our study, treatment of fibroblasts with α-terpineol resulted in a significant increase in phosphor-AKT levels, indicating activation of the PI3K/AKT pathway. Furthermore, inhibition of AKT phosphorylation abolished the induction of TRF1/2 by α-terpineol, highlighting the pivotal role of the PI3K/AKT pathway in mediating the upregulation of TRF1/2 by this agent. These observations suggest that α-terpineol improves shelterin function through modulation of the PI3K/AKT pathway. Taken together, these findings demonstrate the central role of the PI3K/AKT pathway in the regulation of telomere protection, thus pinpointing components of this pathway as novel targets for telomere-based therapies in cancer and age-related diseases.

In summary, our study highlights the crucial roles of TRF1 and TRF2 in maintaining telomere integrity and preventing cellular aging, while also introducing α-terpineol as a promising compound for preserving telomere length and protecting against oxidative DNA damage. The selective protection of nuclear proteins against oxidative damage highlights the importance of understanding the specific antioxidant mechanism(s) of action of each potential activator. To this end, the involvement of the PI3K/AKT pathway in mediating the effects of α-terpineol on shelterin components, suggested here, provides valuable insights into the molecular mechanisms underlying α-terpineol’s putative anti-aging and therapeutic value. Overall, our findings advance our understanding of telomere biology and suggest innovative strategies for mitigating cellular aging and the progression of age-related diseases.

## 5. Patents

The data presented in this study are patent-protected (no. 117-0004899137).

## Figures and Tables

**Figure 1 antioxidants-13-01258-f001:**
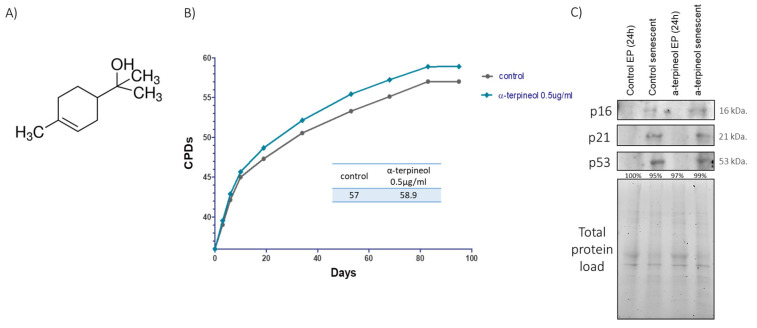
Terpineol extends replicative lifespan. (**A**) Chemical structure of α-terpineol. (**B**) Number of cumulative population doublings (CPDs) of human embryonic fibroblasts continuously treated with 0.5 μg/mL α-terpineol or the diluent DMSO (control) as a function of time in culture. (**C**) Immunoblot analysis of the senescence markers p16, p21 and p53 in cellular extracts from early-passage fibroblasts treated with 0.5 μg/mL α-terpineol or the diluent DMSO (control) for 24 h and terminally senescent fibroblasts grown in media continuously supplemented with 0.5 μg/mL α-terpineol or the diluent DMSO (control), throughout their lifespan. The values of early-passage (T0) control cells were arbitrarily set to 1 or 100% in all described assays and normalization was performed using the total protein load to ensure equal protein loading.

**Figure 2 antioxidants-13-01258-f002:**
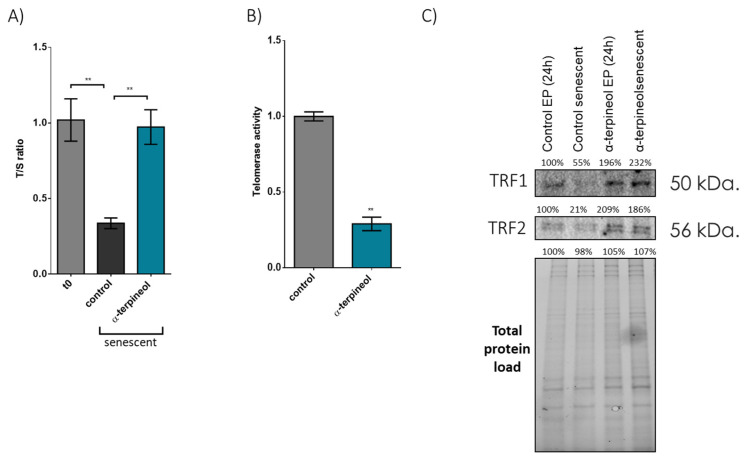
α-Terpineol preserves telomere length during senescence via the shelterin components TRF1 and TRF2: (**A**) Analysis of the indicative telomere length T/S ratio in early-passage (t0) and terminally senescent human fibroblasts grown in media continuously supplemented with 0.5 μg/mL α-terpineol or the diluent DMSO (control), throughout their lifespan. (**B**) Assessment of telomerase activity in human fibroblasts grown in media supplemented with 0.5 μg/mL α-terpineol or the diluent DMSO (control) for 48 h. (**C**) Immunoblot analysis of TRF1 and TRF2 in nuclear extracts from early-passage fibroblasts treated with 0.5 μg/mL α-terpineol or the diluent DMSO (control) for 24 h and terminally senescent fibroblasts grown in media continuously supplemented with 0.5 μg/mL α-terpineol or the diluent DMSO (control), throughout their lifespan. The values of early-passage (T0) control cells were arbitrarily set to 1 or 100% in all described assays and normalization was performed using the total protein load to ensure equal protein loading. **: *p* < 0.01.

**Figure 3 antioxidants-13-01258-f003:**
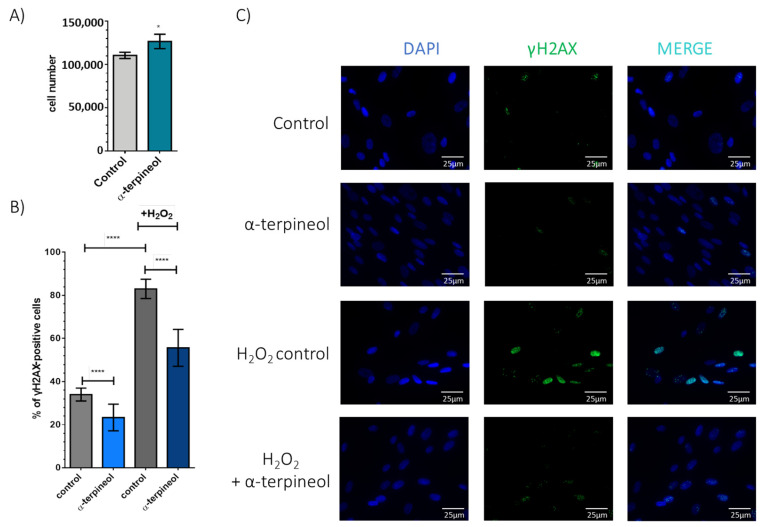
Treatment with terpineol reduces oxidative stress-induced DNA damage and enhances cellular resistance. (**A**) Number of fibroblasts treated with terpineol or DMSO (solvent control) for 24 h following treatment with 300 μM H_2_O_2_ for 2.5 h and a five-day recovery period. (**B**) Quantification of the percentage of γH2AΧ-positive cells and (**C**) representative images of human fibroblasts treated with α-terpineol or DMSO (solvent control) for 24 h following treatment with 300 μM H_2_O_2_ for 2.5 h. γH2AΧ was detected using an anti-γH2AΧ (green) antibody. DNA is visualized using DAPI (4′, 6′-diamidino-2-phenylindole) (blue). Representative images of nuclei and γH2AΧ are shown with or without H_2_O_2_ treatment. ****: *p* < 0.0001, *: *p* < 0.05.

**Figure 4 antioxidants-13-01258-f004:**
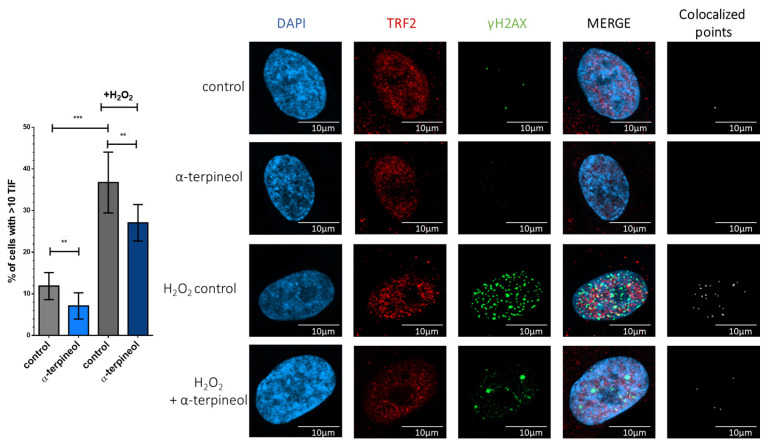
α-Terpineol protects from oxidative stress-induced telomere-specific DNA damage. Telomere dysfunction-induced foci (TIF) assay in human fibroblasts treated with α-terpineol or DMSO (solvent control) for 24 h following treatment with 300 μM H_2_O_2_ for 2.5 h. Colocalization of ΤRF2 and γH2AΧ was determined using anti-TRF2 (red) and anti-γH2AΧ (green) antibodies. DNA was visualized using DAPI (4′, 6′-diamidino-2-phenylindole) (blue). Cells with ten or more TIFs were scored as TIF-positive. Representative images of nuclei, TRF2 and γH2AΧ are shown with or without H_2_O_2_ treatment. ***: *p* < 0.001, **: *p* < 0.01.

**Figure 5 antioxidants-13-01258-f005:**
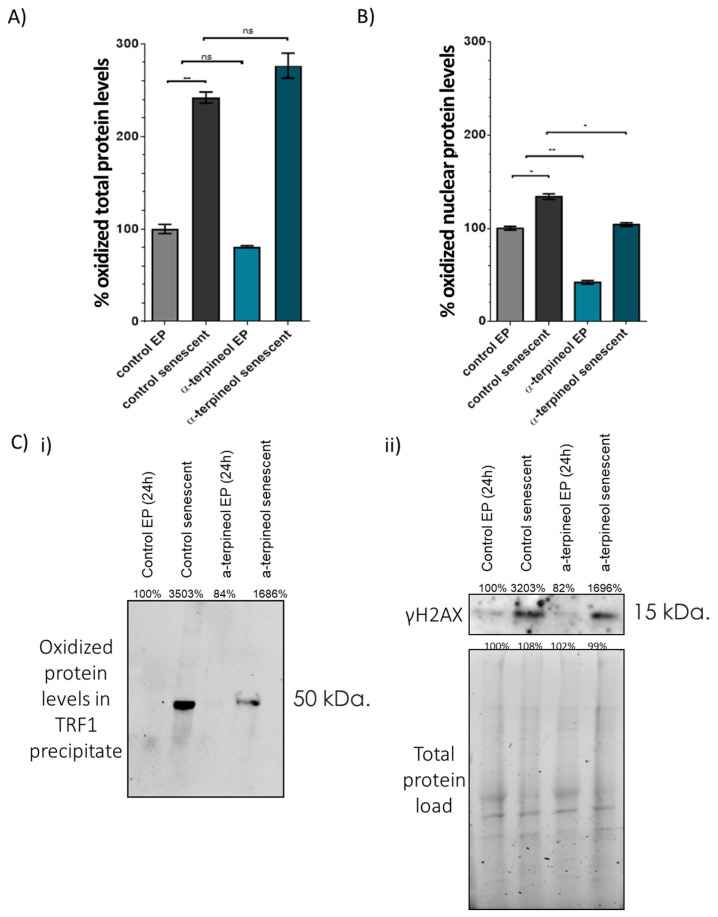
α-Terpineol reduces oxidative damage of nuclear proteins and shelterin components during senescence. % levels of oxidized proteins in (**A**) total and (**B**) nuclear extracts from early-passage fibroblasts treated with 0.5 μg/mL α-terpineol or the diluent DMSO (control) for 24 h and terminally senescent fibroblasts grown in media continuously supplemented with 0.5 μg/mL α-terpineol or the diluent DMSO (control), throughout their lifespan. (**C**) Immunoblot analysis of (**i**) oxidized proteins in TRF1 precipitates and (**ii**) γH2A.X in nuclear extracts from early-passage fibroblasts treated with 0.5 μg/mL α-terpineol or the diluent DMSO (control) for 24 h and terminally senescent fibroblasts grown in media continuously supplemented with 0.5 μg/mL α-terpineol or the diluent DMSO (control), throughout their lifespan. The signal in early-passage (T0) control cells was arbitrarily set to 100% in all described assays and normalization was performed using the total protein load to ensure equal protein loading. **: *p* < 0.01, *: *p* < 0.05.

**Figure 6 antioxidants-13-01258-f006:**
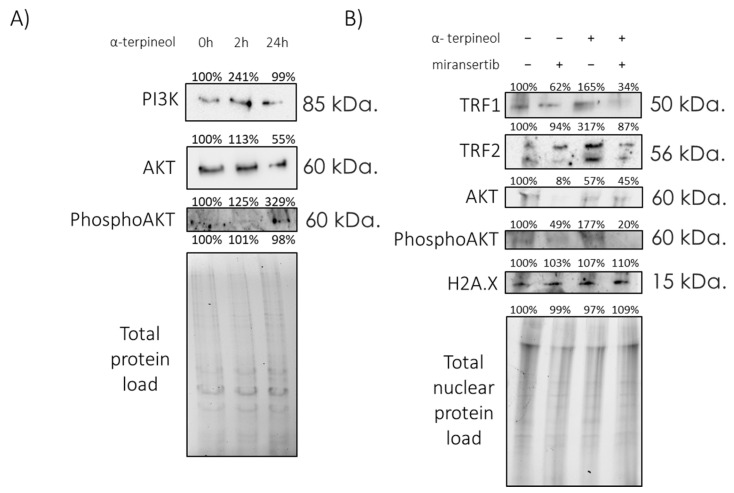
PI3K/AKT signaling mediates the effects of α-terpineol in shelterin components. Immunoblot analysis of the indicated proteins in the (**A**) total protein load of early-passage fibroblasts treated with 0.5 μg/mL α-terpineol for the indicated timepoints and (**B**) nuclear protein extracts from early-passage fibroblasts treated with 0.5 μg/mL α-terpineol or the diluent (DMSO) and the PI3K/AKT inhibitor miransertib for 24 h, as indicated. The signal in control cells was arbitrarily set to 100% in all described assays and normalization was performed using the total protein load to ensure equal protein loading. H2A.X was used as a nuclear protein marker.

## Data Availability

Data contained within the article.

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
