# Peer review of "α-Terpineol Induces Shelterin Components TRF1 and TRF2 to Mitigate Senescence and Telomere Integrity Loss via A Telomerase-Independent Pathway"

_antioxidants, 2024, doi:10.3390/antiox13101258_

Round 1
Reviewer 1 Report
Progressive telomere shortening in dividing somatic cells triggers replicative senescence in culture. Telomeres are protected by the shelterin complex, which includes TRF1 and TRF2, preventing DNA damage at chromosome ends. In this study, the authors identified the monoterpene alpha-terpineol, which enhances TRF1 and TRF2 levels in a telomerase-independent manner, counteracting their senescence-associated decline. Alpha-terpineol also reduced oxidative stress-induced DNA damage in the genome, including telomeric regions, protected oxidation of nuclear proteins including TRF1 during senescence, preserved telomere length, and delayed the onset of senescence. The study linked the PI3K/AKT pathway to the regulation of TRF1 and TRF2 levels by alpha-terpineol. While these findings are intriguing, the quality and clarity of the data require further clarification. Although TRF1 and TRF2 are crucial for telomere capping, their deficiency typically results in telomere dysfunction-induced foci rather than telomere shortening. The evidence supporting alpha-terpineol's specific role in protecting telomeres from oxidative DNA damage remains unconvincing. Additionally, alpha-terpineol appears to negatively regulate telomerase activity, which requires further investigation, as it may influence the fate of telomerase-expressing cells in the context of anti-aging treatments. Specific comments on these points are outlined in detailed comments.
Figure 1A: What is the micromolar equivalent of 0.2-5 μg/ml of alpha-terpineol?
Figure 1B: The CPDs may indicate diminished cellular proliferation in this figure. The authors should evaluate the proportion of proliferating cells (e.g., Ki67-positive cells). Furthermore, to illustrate cellular senescence, they ought to investigate senescence markers, including beta-gal staining, along with the expression of p16, p21, and p53.
Figure 2B: Alpha-terpineol inhibits telomerase activity. Is this impact mediated by Tert or Terc?
Figure 2C: Kindly incorporate the quantification of the TRF1 and TRF2 ratio relative to the total protein load
Figures 3 and 4: Kindly indicate the total number of cells counted. The TRF2 staining quality is poor. Although gamma-H2AX foci in the genome exhibit changes between treated and untreated groups, there is no substantial disparity in the telomeres. Thus, the protective effect of alpha-terpineol on telomeres remains uncertain.
Figure 5: Kindly display the oxidized levels of an alternative nuclear protein as a control.
Figure 6: Kindly incorporate the quantification of the ratio of individual proteins to total protein.
Author Response
Figure 1A: What is the micromolar equivalent of 0.2-5 μg/ml of alpha-terpineol?
We thank the reviewer for his question. The micromolar equivalents for 0.2-5 μg/ml of alpha-terpineol range from 1.2 μM to 32.4 μM. Specifically, 0.5 μg/ml is equivalent to approximately 3.24 μM. We have added this information the respective Results section (lines 220-222)
Figure 1B: The CPDs may indicate diminished cellular proliferation in this figure. The authors should evaluate the proportion of proliferating cells (e.g., Ki67-positive cells). Furthermore, to illustrate cellular senescence, they ought to investigate senescence markers, including beta-gal staining, along with the expression of p16, p21, and p53.
We thank the reviewer for the insightful comment. In response, we have incorporated an immunoblot analysis for the senescence markers p16, p21, and p53 in Figure 1. Unfortunately, as beta-galactosidase and Ki67 staining require fresh cultures, it is not feasible to perform these experiments within the current revision period. Should an extension be granted (given that HDFs take approximately 90 days to become senescent), we would be pleased to include these additional stainings.
Figure 2B: Alpha-terpineol inhibits telomerase activity. Is this impact mediated by Tert or Terc?
We appreciate the reviewer’s question. The TRAPEZE® RT Telomerase Detection Kit (S7710) we used detects telomerase activity primarily through the telomerase reverse transcriptase (Tert) component. Although Terc (telomerase RNA component) is essential for the complex, this kit focuses on the activity related to Tert. The decrease in telomerase activity, coupled with the increase in telomere length observed, suggests that the inhibitory effects of alpha-terpineol are primarily mediated through Tert. We have added this clarification in the discussion section, lines 378-383.
Figure 2C: Kindly incorporate the quantification of the TRF1 and TRF2 ratio relative to the total protein load.
We appreciate the reviewer’s attention to detail. The quantifications presented in the figure have already been normalized to the total protein load. To avoid any confusion, we have revised the figure legends for Figures 1, 2, 5, and 6 to clarify that normalization to total protein load has been performed.
Figures 3 and 4: Kindly indicate the total number of cells counted. The TRF2 staining quality is poor. Although gamma-H2AX foci in the genome exhibit changes between treated and untreated groups, there is no substantial disparity in the telomeres. Thus, the protective effect of alpha-terpineol on telomeres remains uncertain.
For calculations of damaged nuclei at least 100 cells from at least 5 randomly selected fields, including TIF assays. This information has been incorporated in the respective section of Material and Methods (line 203). We confirm that the staining quality of TRF2 is consistent with the manufacturer’s protocol (ab108997, Abcam) and published literature (Nera et al., 2015; Okamoto et al., 2013).
Figure 5: Kindly display the oxidized levels of an alternative nuclear protein as a control.
In response to this valuable comment, we have now added an immunoblot for γH2AX as an additional marker of oxidative damage in the nucleus, in the same samples shown in Figure 5.
Figure 6: Kindly incorporate the quantification of the ratio of individual proteins to total protein.
As previously mentioned, the quantifications of individual proteins presented in this figure have already been normalized to the total protein load. The figure legends for Figures 1, 2, 5, and 6 have been updated to clearly indicate this normalization.
Reviewer 2 Report
The study describes the role of an antioxidant compound, i.e., alpha-terpineol, in the control of oxidative damage and thus, of telomere length and senescence. The information reported are valuable to dissect the specific targets of the compound and the activity of specific proteins in the maintenance of telomere structure and function.
No major critical issues have been identified.
Minor revision could be valuable.
line 3: telomere integrity loss is the cause of senescence, not a consequence as it appears from this title. Alpha-terpineol could appear in the title taking into consideration its role in the effects that have been studied.
line 20: could you include the name of this activator?
lines 20-23: based on this info, the decline associated to senescence is in the level of TRF1 and TRF2. That should be the subject of the title, not telomere loss. It is the activator (alpha-terpineol) that alleviates the sencescence associated phenotype and this should be reported clearly in the title.
lines 47-48: could you change the order of these effects, from DSB repair to apoptosis and senescence following the expected biological timecourse of these events?
lines 85-90: include references supporting the information.
line 120: it should be given in letters because it is a number at the beginning of a sentence.
lines 186-189: include the dilution factors for the antibodies and composition of the dilution buffer.
line 201: is it possible to indicate the final concentration?
line 210: duration of treatment could be recalled?
line 214: could be reported simply as telomere attrition, without "sencescence-associated"? first it is induced telo shortening and then appears senescence.
line 215: how many divisions? from division number X to division number Y, could you please specify?
line 362: one oxidised guanine over how many bases? could you specify?
Author Response
Minor revision could be valuable.
Line 3: Telomere integrity loss is the cause of senescence, not a consequence as it appears from this title. Alpha-terpineol could appear in the title, taking into consideration its role in the effects that have been studied.
We thank the reviewer for this valuable suggestion. We have revised the title to reflect that telomere integrity loss is the cause of senescence and have incorporated alpha-terpineol's role into the title as recommended.
Line 20: Could you include the name of this activator?
As suggested, we have now included the name of the activator in the text.
Lines 20-23: Based on this info, the decline associated with senescence is in the levels of TRF1 and TRF2. That should be the subject of the title, not telomere loss. It is the activator (alpha-terpineol) that alleviates the senescence-associated phenotype, and this should be clearly reported in the title.
We appreciate the reviewer’s suggestion and have revised the title accordingly to emphasize the role of TRF1 and TRF2 levels as well as the role of alpha-terpineol in mitigating the senescence phenotype.
Lines 47-48: Could you change the order of these effects, from DSB repair to apoptosis and senescence, following the expected biological time course of these events?
We thank the reviewer for this insightful comment. We have revised the order of these events in the text to align with the biological sequence.
Lines 85-90: Include references supporting the information.
We thank the reviewer for this note. The relevant references have been added to support the statements in this section.
Line 120: It should be given in letters because it is a number at the beginning of a sentence.
We appreciate the reviewer’s attention to detail. We have now written the number in letters to comply with grammatical conventions.
Lines 186-189: Include the dilution factors for the antibodies and the composition of the dilution buffer.
We thank the reviewer for this important point. The dilution factors for the antibodies and the composition of the dilution buffer have been added to the manuscript (lines 194-195).
Line 201: Is it possible to indicate the final concentration?
The final concentration of 10 μM has been noted in the text to avoid any confusion.
Line 210: Could the duration of treatment be recalled?
We have added the treatment duration (95 days) in the text for clarification.
Line 214: Could this be reported simply as "telomere attrition," without "senescence-associated"? First, it is induced telomere shortening, and then senescence appears.
We have removed the term "senescence-associated" and now refer to this as "telomere attrition" to reflect the correct sequence of events.
Line 215: How many divisions? From division number X to division number Y, could you please specify?
Control cultures underwent 57 cumulative population doublings (CPDs) on average, while the alpha-terpineol-treated cells completed 58.9 CPDs. This information has been included in the text.
Line 362: One oxidized guanine over how many bases? Could you specify?
According to the work by Opresko et al. (2005), a single 8-oxo-guanine lesion within a telomeric repeat reduces the binding of TRF1 and TRF2 by over 50% compared to undamaged telomeric DNA. We have added this information, in the text to enhance clarity (lines 386-389).
According to the same research, more dramatic reductions in TRF1 and TRF2 binding were observed with multiple 8-oxo-guanine lesions in tandem repeats. Additionally, the presence of intermediates from base excision repair, such as abasic sites or single nucleotide gaps, further disrupted the binding of TRF proteins to the telomeric DNA.
Round 2
Reviewer 1 Report
The authors have addressed my concerns.
I have no further questions.